# Pharmacometrics of high-dose ivermectin in early COVID-19 from an open label, randomized, controlled adaptive platform trial (PLATCOV)

William HK Schilling[1,2]*[†], Podjanee Jittamala[1,3†], James A Watson[1,2], Maneerat Ekkapongpisit[1], Tanaya Siripoon[4], Thundon Ngamprasertchai[4], Viravarn Luvira[4], Sasithorn Pongwilai[1], Cintia Cruz[1,2], James J Callery[1,2], Simon Boyd[1,2], Varaporn Kruabkontho[1], Thatsanun Ngernseng[1], Jaruwan Tubprasert[1], Mohammad Yazid Abdad[1,2], Nattaporn Piaraksa[1], Kanokon Suwannasin[1], Pongtorn Hanboonkunupakarn[5], Borimas Hanboonkunupakarn[1,4], Sakol Sookprome[5], Kittiyod Poovorawan[1,4], Janjira Thaipadungpanit[1,4], Stuart Blacksell[1,2], Mallika Imwong[1,6], Joel Tarning[1,2], Walter RJ Taylor[1,2], Vasin Chotivanich[7], Chunlanee Sangketchon[8], Wiroj Ruksakul[7], Kesinee Chotivanich[1,4], Mauro Martins Teixeira[9], Sasithon Pukrittayakamee[1,4], Arjen M Dondorp[1,2], Nicholas PJ Day[1,2], Watcharapong Piyaphanee[4], Weerapong Phumratanaprapin[4], Nicholas J White[1,2]*, on behalf of the PLATCOV Collaborative Group

[1]Mahidol Oxford Tropical Medicine Research Unit, Faculty of Tropical Medicine, Mahidol University, Bangkok, Thailand; [2]Centre for Tropical Medicine and Global Health, Nuffield Department of Medicine, Oxford University, Oxford, United Kingdom; [3]Department of Tropical Hygiene, Faculty of Tropical Medicine, Mahidol University, Bangkok, Thailand; [4]Department of Clinical Tropical Medicine, Faculty of Tropical Medicine, Mahidol University, Bangkok, Thailand; [5]Bangplee Hospital, Ministry of Public Health, Bangkok, Thailand; [6]Department of Molecular Tropical Medicine and Genetics, Faculty of Tropical Medicine, Mahidol University, Bangkok, Thailand; [7]Faculty of Medicine, Navamindradhiraj University, Bangkok, Thailand; [8]Faculty of Science and Health Technology, Navamindradhiraj University, Bangkok, Thailand; [9]Department of Biochemistry and Immunology, Universidade Federal de Minas Gerais, Belo Horizonte, Brazil

*For correspondence:
william@tropmedres.ac (WHKS);
nickw@tropmedres.ac (NJW)

†These authors contributed equally to this work

## Abstract

**Background:** There is no generally accepted methodology for in vivo assessment of antiviral activity in SARS-CoV-2 infections. Ivermectin has been recommended widely as a treatment of COVID-19, but whether it has clinically significant antiviral activity in vivo is uncertain.

**Methods:** In a multicentre open label, randomized, controlled adaptive platform trial, adult patients with early symptomatic COVID-19 were randomized to one of six treatment arms including high-dose oral ivermectin (600 µg/kg daily for 7 days), the monoclonal antibodies casirivimab and imdevimab (600 mg/600 mg), and no study drug. The primary outcome was the comparison of viral clearance rates in the modified intention-to-treat population. This was derived from daily $\log_{10}$ viral densities in standardized duplicate oropharyngeal swab eluates. This ongoing trial is registered at https://clinicaltrials.gov/ (NCT05041907).

**Results:** Randomization to the ivermectin arm was stopped after enrolling 205 patients into all arms, as the prespecified futility threshold was reached. Following ivermectin, the mean estimated rate of SARS-CoV-2 viral clearance was 9.1% slower (95% confidence interval [CI] –27.2% to +11.8%; n=45) than in the no drug arm (n=41), whereas in a preliminary analysis of the casirivimab/imdevimab arm it was 52.3% faster (95% CI +7.0% to +115.1%; n=10 (Delta variant) vs. n=41).

**Conclusions:** High-dose ivermectin did not have measurable antiviral activity in early symptomatic COVID-19. Pharmacometric evaluation of viral clearance rate from frequent serial oropharyngeal qPCR viral density estimates is a highly efficient and well-tolerated method of assessing SARS-CoV-2 antiviral therapeutics in vivo.

**Funding:** 'Finding treatments for COVID-19: A phase 2 multi-centre adaptive platform trial to assess antiviral pharmacodynamics in early symptomatic COVID-19 (PLAT-COV)' is supported by the Wellcome Trust Grant ref: 223195/Z/21/Z through the COVID-19 Therapeutics Accelerator.
**Clinical trial number:** NCT05041907.

## Editor's evaluation

This valuable clinical trial demonstrated in a convincing fashion that ivermectin does not increase the rate of SARS-CoV-2 clearance from the oral compartment. This work will be of interest to clinicians and virologists and further demonstrates the use of viral clearance rate as a possible surrogate marker for SARS-CoV-2 antiviral trials.

## Introduction

Effective, safe, well-tolerated, and inexpensive oral antiviral agents are needed for the early treatment of COVID-19. Monoclonal antibodies, mainly directed against the SARS-CoV-2 spike protein, have proved effective in preventing and treating COVID-19 (*Weinreich et al., 2021*; *O'Brien et al., 2021*), but they are expensive, require parenteral administration, and are very vulnerable to the emergence of spike protein mutations (*Bruel et al., 2022*). Recently, large randomized controlled trials have shown clinical efficacy in the treatment of early COVID-19 for the ribonucleoside analog molnupiravir and the protease inhibitor nirmatrelvir (in combination with ritonavir) (*Jayk Bernal et al., 2022*; *Hammond et al., 2022*), but these drugs are not yet widely available, especially in low- and middle-income settings. There have been no reported randomized comparisons between these expensive medicines. In the absence of comparative assessments, and uncertainty over antiviral efficacy, national treatment guidelines vary widely across the world.

Early in the COVID-19 pandemic, considerable attention was focussed on available drugs that might have useful antiviral activity (*Robinson et al., 2022*). Notable and widely promoted repurposing candidates included hydroxychloroquine, remdesivir, and ivermectin. The macrocyclic lactone endectocide ivermectin was pursued after a laboratory study suggested antiviral activity against SARS-CoV-2 (*Caly et al., 2020*). This in vitro activity, extensive experience in mass treatments for onchocerciasis, a well-established safety profile, and claims of clinical benefit, led to ivermectin being added to COVID-19 treatment guidelines in many countries, particularly in Latin America (*Mega, 2020*). Several small clinical trials have reported a survival benefit for ivermectin, although the quality of these trials has been questioned (*Lawrence et al., 2021*). Ivermectin's relatively weak in vitro activity in relation to achievable blood levels in vivo has argued for the evaluation of maximum tolerated doses (c.600 µg/kg/day). The large TOGETHER platform trial excluded substantial clinical benefit with ivermectin in early COVID-19 infection. This was evaluated using a composite outcome of hospitalization or lengthy (> 6 hr) emergency department visit (relative risk, 0.90; 95% Bayesian credible interval, 0.70–1.16), but the TOGETHER trial used a relatively low dose of ivermectin; 400 µg/kg/day for only 3 days (*Reis et al., 2022*).

To resolve the uncertainty over efficacy, we measured the in vivo antiviral activity of high-dose ivermectin in previously healthy adults with early symptomatic COVID-19 infection.

## Methods

PLATCOV is an ongoing open label, randomized, controlled adaptive platform trial designed to provide a standardized quantitative comparative method for in vivo assessment of potential antiviral treatments in early symptomatic COVID-19. The primary outcome is the change in the rate of viral clearance compared with the contemporaneous no study drug arm. This is measured as the change in the slope of the $\log_{10}$ oropharyngeal viral clearance curve (*Watson et al., 2022*). The trial was conducted in the Hospital for Tropical Diseases, Faculty of Tropical Medicine, Mahidol University, Bangkok; Bangplee hospital, Samut Prakarn; and Vajira hospital, Navamindradhiraj University, Bangkok, all in Thailand (see Notes). The trial was approved by local and national research ethics boards in Thailand (Faculty of Tropical Medicine Ethics Committee, Mahidol University, FTMEC Ref: TMEC 21-058) and the Central Research Ethics Committee (CREC, Bangkok, Thailand, CREC Ref: CREC048/64BP-MED34) and by the Oxford University Tropical Research Ethics Committee (OxTREC EC, Oxford, UK, OxTREC Ref: 24-21). All patients provided fully informed written consent. The trial was coordinated and monitored by the Mahidol Oxford Tropical Medicine Research Unit (MORU). The PLATCOV trial was overseen by a trial steering committee and results were reviewed by a data and safety monitoring board (DSMB). The funders had no role in the design, conduct, analysis, or interpretation of the trial. The ongoing trial is registered at https://clinicaltrials.gov/ (NCT05041907).

**Table 1.** Ivermectin dosing table.
Daily dose of ivermectin given to patients based on weight.

| Weight in kg | Number of 6 mg tablets | Dose in mg | Dose in mg/kg |
|---|---|---|---|
| 40 to <50 | 4 | 24 | 0.49–0.6 |
| 50 to <60 | 5 | 30 | 0.51–0.6 |
| 60 to <70 | 6 | 36 | 0.52–0.6 |
| 70 to <80 | 7 | 42 | 0.53–0.6 |
| 80 to <90 | 8 | 48 | 0.54–0.6 |
| 90 to <100 | 9 | 54 | 0.55–0.6 |
| ≥100 | 10 | 60 | ≤0.6 |

### Participants

Patients presenting to the Acute Respiratory Infections outpatient clinics for COVID-19 testing were prescreened for study eligibility. Previously healthy adults aged between 18 and 50 years were eligible for the trial if they had early symptomatic COVID-19 (i.e., reported symptoms for not more than 4 days), oxygen saturation ≥96%, were unimpeded in activities of daily living, and were willing to give fully informed consent and adhere to the study protocol. SARS-CoV-2 positivity was defined either as a nasal lateral flow antigen test which became positive within 2 min (STANDARD Q COVID-19 Ag Test, SD Biosensor, Suwon-si, Republic of Korea) or a positive PCR test within the previous 24 hr with a cycle threshold value (Ct) <25 (all viral gene targets), both suggesting high viral loads. The latter was added on November 25, 2021 to include those patients with recent PCR results confirming high viral loads. This was the only change to the pretrial prespecified inclusion/exclusion criteria. Exclusion criteria included taking any potential antivirals or preexisting concomitant medications, chronic illness or significant comorbidity, hematological or biochemical abnormalities, pregnancy (a urinary pregnancy test was performed in females), breastfeeding, or contraindication or known hypersensitivity to any of the study drugs.

### Randomization and interventions

Randomization was performed via a centralized web-app designed by MORU software engineers using RShiny, hosted on a MORU webserver. For all sites, envelopes were generated initially as backup. The no study drug arm comprised a minimum proportion of 20% and uniform randomization ratios were then applied across the treatment arms. For example, for five intervention arms plus the no study drug arm, 20% of patients would be randomized to no study drug and 16% to each of the five interventions. Additional details on the randomization are provided in Appendix 3. All patients received standard symptomatic treatment.

Ivermectin (600 µg/kg; 6 mg tablets; Atlantic Laboratories, Thailand) was given once daily for 7 days with food (see *Table 1* below). Patients were supplied with a hospital meal of 500–600 kcal containing 20–25% fat. Casirivimab/imdevimab (600 mg/600 mg; Roche, Switzerland) was given once by intravenous infusion following randomization. During this period, other patients were randomized to remdesivir, favipiravir, or fluoxetine (added to the randomization list April 1, 2022).

## Trial procedures

Eligible patients were admitted to the study ward. Baseline investigations included a full clinical examination, rapid SARS-CoV-2 antibody test (BIOSYNEX COVID-19 BSS IgM/IgG, Illkirch-Graffenstaden, France), blood sampling for hematology and biochemistry, an electrocardiogram and a chest radiograph (following local guidance, but this was not a study requirement). After randomization, oropharyngeal swabs (two swabs from each tonsil) were taken as follows. A Thermo Fisher MicroTest flocked swab was rotated against the tonsil through 360° four times and placed in Thermo Fisher M4RT viral transport medium (3 mL). On subsequent days (days 1–7 and then after discharge on day 14), a single swab was taken from each tonsil (left and right, total of 2 swabs). Swabs were transferred separately at 4–8°C, and then frozen at –80°C within 48 hr. Thus, each patient had a total of 20 swabs.

The TaqCheck SARS-CoV-2 Fast PCR Assay (Applied Biosystems, Thermo Fisher Scientific, Waltham, MA) was used to quantitate viral load (RNA copies per mL). This assay is a multiplexed real-time PCR method, which detects the SARS-CoV-2 N-gene and S-gene as well as human RNase P in a single reaction. RNase P was used to correct for variation in the sample human cell content. The viral load was quantified against known standards using the ATCC heat-inactivated SARS-CoV-2 VR-1986HK strain 2019-nCoV/USA-WA1/2020. Viral genetic variants were identified using real-time PCR genotyping with the TaqMan SARS-CoV-2 Mutation Panel. Plasma ivermectin concentrations were determined on days 3 and 7 using validated high-performance liquid chromatography linked with tandem mass spectrometry (*Tipthara et al., 2021*; *Kobylinski et al., 2020*). Adverse events were graded according to the Common Terminology Criteria for Adverse Events v.5.0 (CTCAE). Adverse event summaries were generated if the adverse event was grade 3 or higher and the adverse event was new, or increased in intensity from study drug administration until the end of the follow-up period. Serious adverse events were recorded separately and reported to the DSMB.

## Outcome measures, stopping rules, and statistical analysis

The primary measure was the rate of viral clearance, expressed as a slope coefficient and presented as a half-life. This was estimated under a Bayesian hierarchical linear model fitted to the daily $\log_{10}$ viral load measurements between days 0 and 7 (18 measurements per patient. Each swab value was treated as independent and identically distributed conditional on the model). Viral loads below the lower limit of quantification (Ct values>40) were treated as left-censored under the model with a known censoring value. The PCRs were done on 96-well plates, each of which included 12 standards of known viral density. The Ct values from the patient swabs were then converted to copies per mL under standard curves estimated using the control data from all available plates (a mixed-effects linear regression model with a random slope and intercept for each plate; each plate therefore has a slightly different left-censoring value). The main model used weakly informative priors (see Appendix 4). The viral clearance rate (i.e., slope coefficient from the model fit) is inversely proportional to the clearance half-life ($t_{1/2}=\log_{10} 0.5/\text{slope}$). The treatment effect is defined as the percentage change in the viral clearance rate relative to the contemporaneous no study drug arm (i.e., how much the treatment accelerates viral clearance) (*Watson et al., 2022*). A 50% increase in viral clearance rate is thus equal to a 33% reduction in the viral clearance half-life. All-cause hospitalization for clinical deterioration (until day 28) was a secondary endpoint. For each studied intervention, the sample size is adaptive.

Under the linear model, for each intervention, the treatment effect β is encoded as a multiplicative term on the time since randomization: $e^{\beta T}$, where T=1, if the patient was assigned the intervention, and 0 otherwise. Under this specification, β=0 implies no effect (no change in slope), and β>0 implies increase in slope relative to the population mean slope. Stopping rules are then defined with respect to the posterior distribution of β, with futility defined as Prob[β<$\lambda$]>0.9; and success defined as Prob[β>$\lambda$]>0.9, where $\lambda$≥0. Larger values of $\lambda$ imply smaller sample size to stop for futility but a larger sample size to stop for efficacy. $\lambda$ was chosen so that it would result in reasonable sample size requirements, as determined previously using a simulation approach based on modeled serial viral load data (*Watson et al., 2022*). This modelling work suggested that a value of $\lambda$=log(1.05) (i.e., 5% increase) would require approximately 50 patients to demonstrate increases in the rate of viral clearance of ~50%, with control of both type 1 and type 2 errors at 10%. The first interim analysis (n=50) was prespecified as unblinded in order to review the methodology and the stopping rules (notably the value of $\lambda$). Following this, the stopping threshold was increased from 5% to 12.5% ($\lambda$=log(1.125)) because the treatment effect of casirivimab/imdevimab against the SARS-CoV-2 Delta variant was

larger than had been expected and the estimated residual error was greater than previously estimated. Thereafter trial investigators were blinded to the virus clearance results. Interim analyses were planned for every batch of additional 25 patients' PCR data, however, because of delays in setting up the PCR analysis pipeline, the second interim analysis was delayed until April 2022. By that time, data from 145 patients were available (29 patients randomized to ivermectin and 26 patients randomized to no study drug).

All analyses were done in a modified intention-to-treat (mITT) population, comprising patients who had >3 days follow-up qPCR data. The safety population includes all patients who received at least one dose of the intervention. A series of linear and nonlinear Bayesian hierarchical models were fitted to the serial viral quantitative PCR (qPCR) data (Appendix 4). For the pharmacokinetic analysis, a previously developed two-compartment disposition model with four transit compartments adjusted for body weight was fitted to the plasma ivermectin levels (*Kobylinski et al., 2020*). Drug exposures were summarized as the areas under the plasma concentration time curves until 72 hr ($AUC_{0-72}$) and the maximum peak concentrations ($C_{max}$).

This report describes the ivermectin results compared with no treatment, and also includes the unblinded results for the first 10 patients who received casirivimab/imdevimab (recruited until December 16, 2021) to illustrate the pharmacometric method's sensitivity. All data analysis was done in R version 4.0.2. Model fitting was done in *stan* via the *rstan* interface. All code and data are openly accessible via GitHub: https://github.com/jwatowatson/PLATCOV-Ivermectin (*Schilling, 2023*; copy archived at swh:1:rev:9cd724266ab4f7eab09c5d2a7fd50dac52782957).

## Results

The trial began recruitment on September 30, 2021. On April 18, 2022, ivermectin enrolment was stopped as the prespecified futility margin had been reached. Of the 274 patients screened by then (*Figure 1*), 224 had been randomized to either ivermectin (46 patients), casirivimab/imdevimab (40 patients; only the unblinded first 10 are reported here), no study drug (45 patients), or other interventions (93 patients: remdesivir, favipiravir, and fluoxetine). This analysis data set therefore comprised 101 patients (46 ivermectin, 10 casirivimab/imdevimab, and 45 no study drug), of whom 5 patients were excluded for either changing treatment before day 2 (n=3), withdrawing from the study (n=1), or because there was no detectable viral RNA at all timepoints (n=1) (*Figure 1*). In the mITT population (n=96), 60% were female, the median age was 27 (interquartile range [IQR] 25–31) years and the median duration of illness at enrolment was 2 (IQR 2–3) days. Overall, 95% of patients had received at least one dose of a COVID-19 vaccine (*Table 2*). The median (range) daily ivermectin dose was 550 µg/kg/day (490–600 µg/kg/day). All patients recovered without complications. Virus variants spanned Delta (B.1.617.2), prevalent when the study began, then Omicron BA.1 (B.1.1.529), and then Omicron BA.2 (B.1.1.529) (*Figure 1—figure supplement 1*).

### Virological responses

Ninety-six patients in the mITT population had a median of 18 viral load measurements each between days 0 and 7 (range 8–18), of which 7% (121/1700) were below the lower limit of detection. The baseline geometric mean oropharyngeal viral load was $3.6×10^5$ RNA copies/mL (IQR $7.8×10^4$ to $2.8×10^6$) (*Figure 2a*). Oropharyngeal viral loads declined substantially faster in casirivimab/imdevimab recipients compared to both the ivermectin and no study drug arms (*Figure 2b*; *Figure 3*). Under a Bayesian hierarchical linear model, the population mean viral clearance half-life was estimated to be 19.2 hr (95% CI 14.8–23.9 hr) for the no study drug arm. Relative to the no study drug arm, clearance of oropharyngeal virus in patients randomized to ivermectin was 9.1% slower (95% CI –27.2% to +11.8%), whereas with casirivimab/imdevimab it was 52.3% faster (95% CI +7.0% to +115.1%) (*Figure 4*). This corresponded to prolongation of virus clearance half-life by 1.9 hr (95% CI –2.1 to +6.6) for ivermectin and shortening by 6.5 hr (95% CI –12.0 to –1.1) for casirivimab/imdevimab (*Figure 5*). In the no study drug arm, there was considerable inter-individual variability in viral clearance; mean estimated half-life values varied from 7 to 42 hr (*Figure 5*, *Figure 5—figure supplement 1*).

Targeted viral genotyping (Appendix 1) indicated that all 10 casirivimab/imdevimab recipients had the SARS CoV-2 Delta variant (B.1.617.1) (*Figure 5*). The slope and intercept in all models were adjusted for site and virus variant. There were no apparent differences in virus clearance rates across

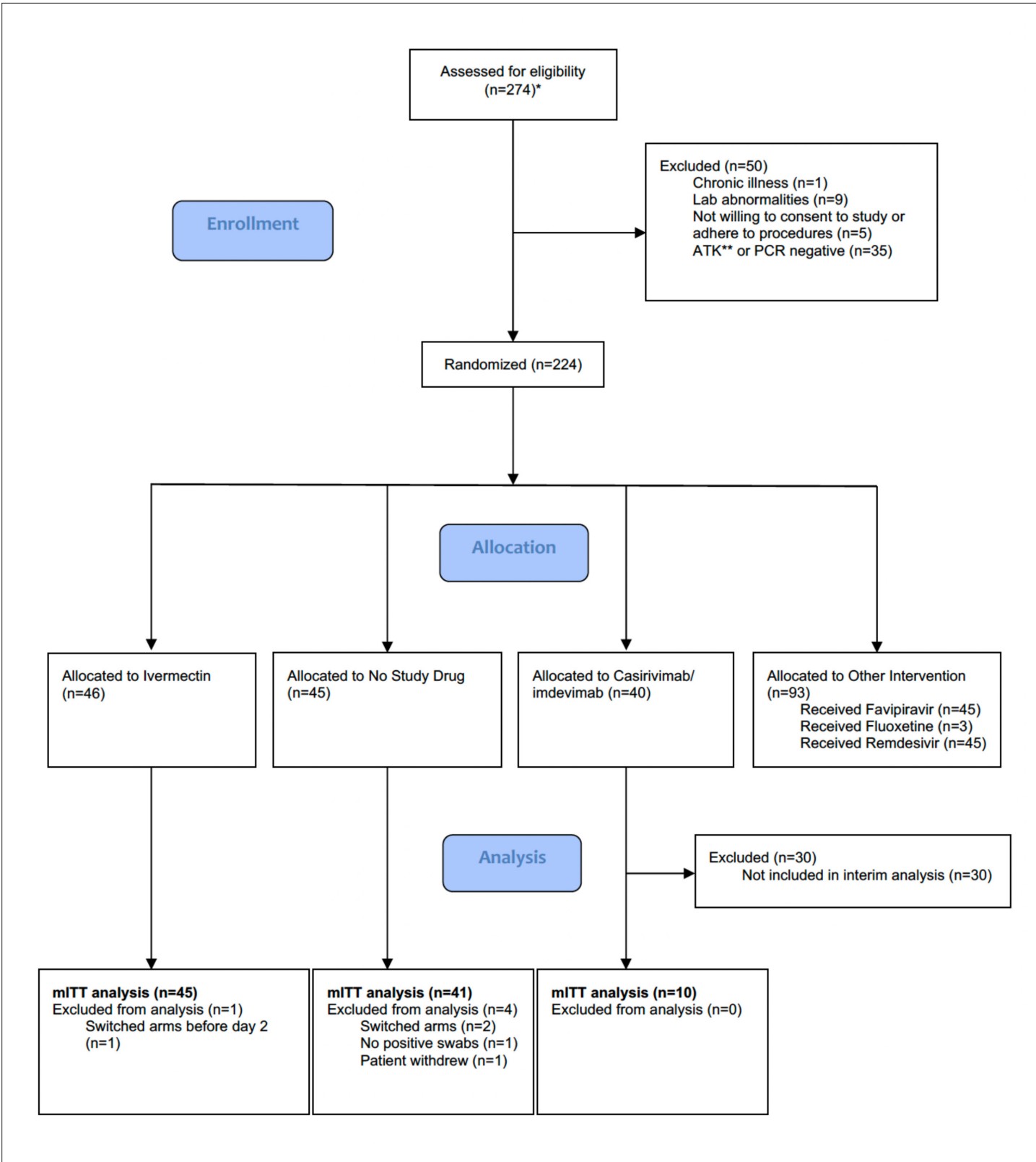

**Figure 1.** Summary of patient characteristics included in the mITT population (n=96). Study CONSORT diagram for the ivermectin analysis. *Prescreening occurred in the hospitals' Acute Respiratory Infection (ARI) units. Potentially eligible participants (based on age, duration of symptoms, reported comorbidities, and a willingness to consider study participation) were selected by the ARI Nurses to be contacted by the study team. As a

*Figure 1 continued on next page*

*Figure 1 continued*

result, a high proportion of those assessed for eligibility participated in the study. **SARS CoV-2 Antigen Test Kit (STANDARD Q COVID-19 Ag Test, SD Biosensor, Suwon-si, Republic of Korea). mITT, modified intention-to-treat.

The online version of this article includes the following figure supplement(s) for figure 1:

**Figure supplement 1.** Randomization dates and virus variants of all patients with available PCR data (n=99).

the different virus variants/subvariants; however, relative to the Delta variant, patients with the Omicron BA.2 subvariant had higher baseline viral loads (3.6-fold higher, 95% CI 1.4–9.4), and patients with the Omicron BA.1 subvariant had lower baseline viral loads (0.4-fold lower, 95% CI 0.1–1.1) (*Figure 4—figure supplements 1–2*).

All analytical models of oropharyngeal virus clearance were in excellent agreement, giving near identical point estimates and credible intervals (*Figure 4—figure supplement 3*). The best fit was the nonlinear model which allows some patients to have viral load increases after randomization (i.e., enrolment before reaching peak viral load), followed by a log-linear decrease. There was no relationship between viral clearance rates and the ivermectin plasma $AUC_{0-72}$ (p=0.8) or $C_{max}$ (p=0.9). Drug exposures were high: all patients had significantly higher plasma concentrations than predicted under the pharmacokinetic model fitted to healthy volunteer data (relative bioavailability 2.6; *Figure 3*; *Kobylinski et al., 2020*).

## Adverse effects

The oropharyngeal swabbing and all treatments were well-tolerated. The three serious adverse events were all in the no study drug arm (see *Supplementary file 1*; *Supplementary file 2*). Two patients had raised creatinine phosphokinase (CPK) levels (>10 times ULN) attributed to COVID-19-related skeletal muscle damage. This improved with fluids and supportive management. The third patient was readmitted 1 day after discharge because of chest pain and lethargy. All investigations were normal and the patient was discharged the following day. Six patients reported transient visual disturbance after taking ivermectin (although not classified as grade 3 or above). Three of these withdrew from the treatment (Appendix 2). All visual symptoms resolved quickly after the drug was stopped. Ophthalmology review confirmed that no visual abnormality remained.

## Discussion

These first data from the PLATCOV adaptive platform study show that ivermectin does not have a measurable antiviral effect in early symptomatic COVID-19 under our study methodology. In contrast, the preliminary results with casirivimab/imdevimab in patients infected with the SARS-CoV-2 Delta variant showed an approximate 50% acceleration in the viral clearance rate. This confirmed that the study methodology identifies efficiently those treatments which have clinically relevant antiviral effects in vivo. It remains uncertain whether any of the proposed, and often recommended, repurposed potential antiviral treatments have significant in vivo antiviral activity in COVID-19. This continued uncertainty, after 3 years of the pandemic, highlights the limitations of the tools currently used to assess antiviral activity in vivo. Clinically effective monoclonal antibodies and specific antiviral drugs

**Table 2.** Summary of patient characteristics included in the mITT population (n=96).

| Treatment arm | Number (total n=96) | Age median, years (range) | Baseline viral load mean log$_{10}$ copies per mL (range) | Vaccine doses received previously median (range) | Antibody positive at baseline from rapid test (%)* | Male (%) | Sites HTD (n=87) | BP (n=5) | VJ (n=4) |
|---|---|---|---|---|---|---|---|---|---|
| Casirivimab/ imdevimab | 10 | 26.5 (18–31) | 5.5 (3.7–7.8) | 2 (0–3) | 50 | 20 | 10 | 0 | 0 |
| Ivermectin | 45 | 29 (19–45) | 5.7 (1.9–7.6) | 2 (0–4) | 78 | 47 | 41 | 2 | 2 |
| No study drug | 41 | 27 (20–43) | 5.5 (3–7.7) | 2 (2–4) | 90 | 44 | 36 | 3 | 2 |

*Defined as IgM or IgG present at enrolment on the rapid antibody test (BIOSYNEX COVID-19 BSS IgM/IgG, Illkirch-Graffenstaden, France) used as per manufacturer's instructions.
HTD: Hospital for Tropical Diseases. BP: Bangplee Hospital. VJ: Vajira Hospital.

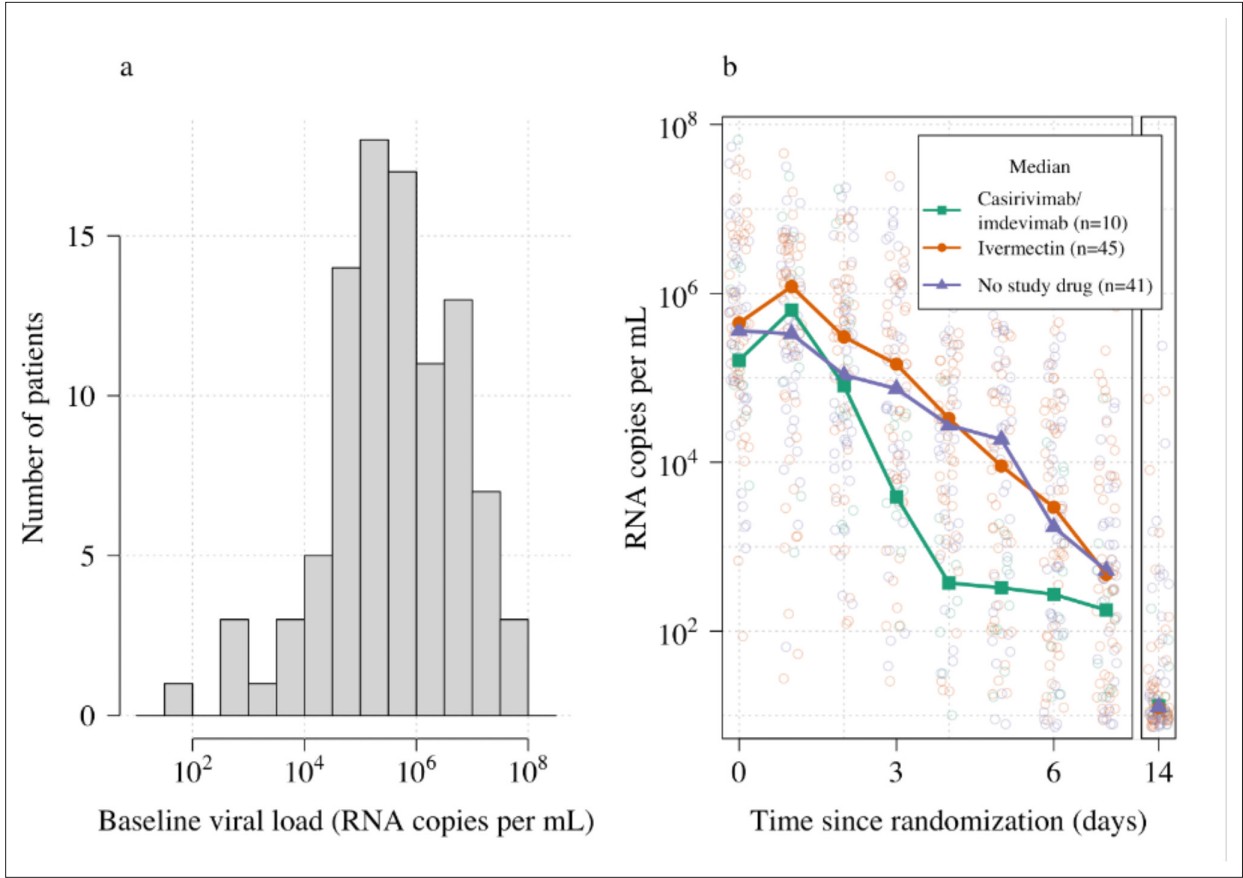

**Figure 2.** Summary of oropharyngeal viral load data in the analysis data set (n=96). (a) Distribution of viral loads at randomization (median of 4 swabs per patient). (b) Individual serial viral load data with x-axis jitter. Median values by study arm are overlaid. The day 14 samples are not used in the primary analysis.

have been developed. It is increasingly accepted that they are most effective early in the course of COVID-19 infection (*O'Brien et al., 2021*; *Jayk Bernal et al., 2022*; *Hammond et al., 2022*; *Gottlieb et al., 2022*) (whereas anti-inflammatory agents have life-saving benefit later in the disease process when severe pneumonitis has developed). Unfortunately, these efficacious antiviral medicines are not generally available outside high-income settings. Meanwhile repurposed therapeutics, which offered the prospect of affordable and generally available medicines, have been selected for clinical use based on in vitro activity in cell cultures, sometimes on animal studies, or based upon clinical trials which often had subjective or infrequent endpoints, or were conducted in late-stage infections in hospitalized patients. The majority of these trials have been underpowered. In a review of 1314 registered COVID-19 studies, of which 1043 (79%) were randomized controlled trials, the median (IQR) sample size was 140 patients (70–383) (*McLean et al., 2023*). These uncertainties have created confusion and, for ivermectin, strongly polarized views.

The method of assessing antiviral activity in early COVID-19 reported here builds on extensive experience of antiviral pharmacodynamic assessments in other viral infections. It has the advantage of simplicity. It also avoids many of the limitations of unvalidated in vivo animal models (*Muñoz-Fontela et al., 2022*). Only a relatively small number of patients are needed to identify antiviral activity in vivo (*Watson et al., 2022*). In this trial, with only 41 controls and 45 subjects receiving ivermectin, acceleration in viral clearance of more than 12.5% could be excluded with high certainty. Smaller numbers are required to show efficacy. These sample sizes are an order of magnitude smaller than required for the more commonly used endpoint of time to viral clearance (PCR negativity) (*Watson et al., 2022*). In addition, the procedures are well-tolerated: daily oropharyngeal swabbing is much more acceptable than frequent nasopharyngeal sampling. Oropharyngeal viral loads have been shown to be both more and less sensitive for the detection of SARS-CoV-2 infection. Although rates of viral clearance are

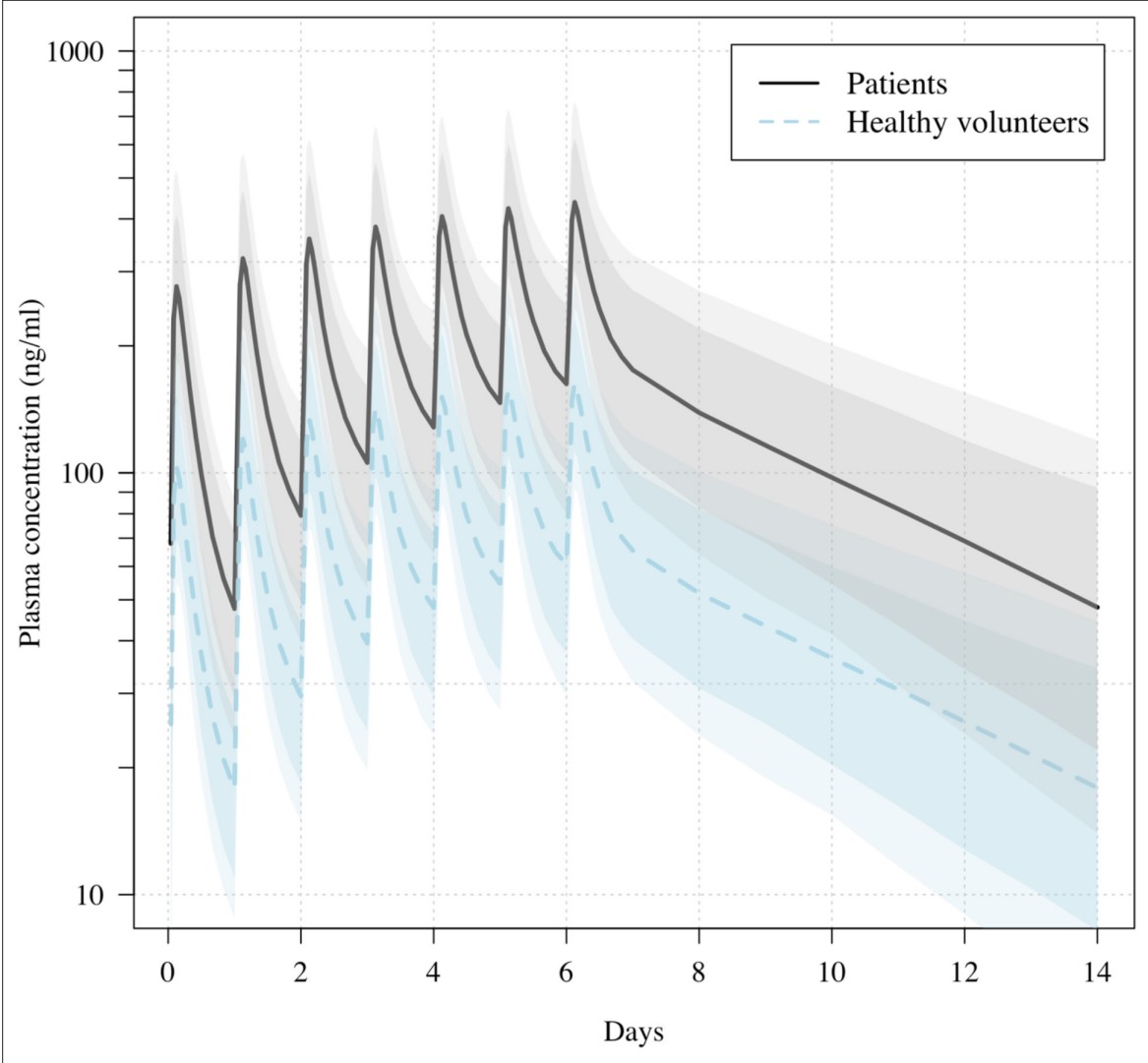

**Figure 3.** Predicted ivermectin plasma concentrations over time under the population PK model fit to data from healthy volunteers (*Kobylinski et al., 2020*), and the ivermectin patients in the PLATCOV study. Mean predicted concentrations with 80% and 95% confidence intervals are shown for daily dosing of 600 μg/kg ivermectin in a 70 kg adult over 1 week for patients (thick line) and healthy volunteers (dashed line). The mean relative bioavailability in patients compared to healthy volunteers was estimated as 2.6.

very likely to be similar from the two body sites, this should be established for comparison with other studies. Using less frequent nasopharyngeal sampling in larger numbers of patients, clinical trials of monoclonal antibodies, molnupiravir, and ritonavir-boosted nirmatrelvir, have each shown that accelerated viral clearance is associated with improved clinical outcomes (*Weinreich et al., 2021*; *Jayk Bernal et al., 2022*; *Hammond et al., 2022*). These data suggest that reduction in viral load could be used as a surrogate of clinical outcome in COVID-19. In contrast, the PINETREE study, which showed that remdesivir significantly reduced disease progression in COVID-19, did not find an association between viral clearance and therapeutic benefit. This seemed to refute the usefulness of viral clearance rates as a surrogate for rates of clinical recovery. However, the infrequent sampling in all these studies substantially reduced the precision of the viral clearance estimates (and thus increased the risk of type 2 errors). Using the frequent sampling employed in the PLATCOV study, we have shown recently that remdesivir does accelerate SARS-CoV-2 viral clearance (*Jittamala et al., 2023*), as would be expected from an efficacious antiviral drug. This is consistent with therapeutic responses in other viral infections (*Natori et al., 2018*; *Mellors et al., 1997*). Taken together, the weight of evidence suggests that accelerated viral clearance does reflect therapeutic efficacy in early COVID-19, although more information will be required to characterize this relationship adequately.

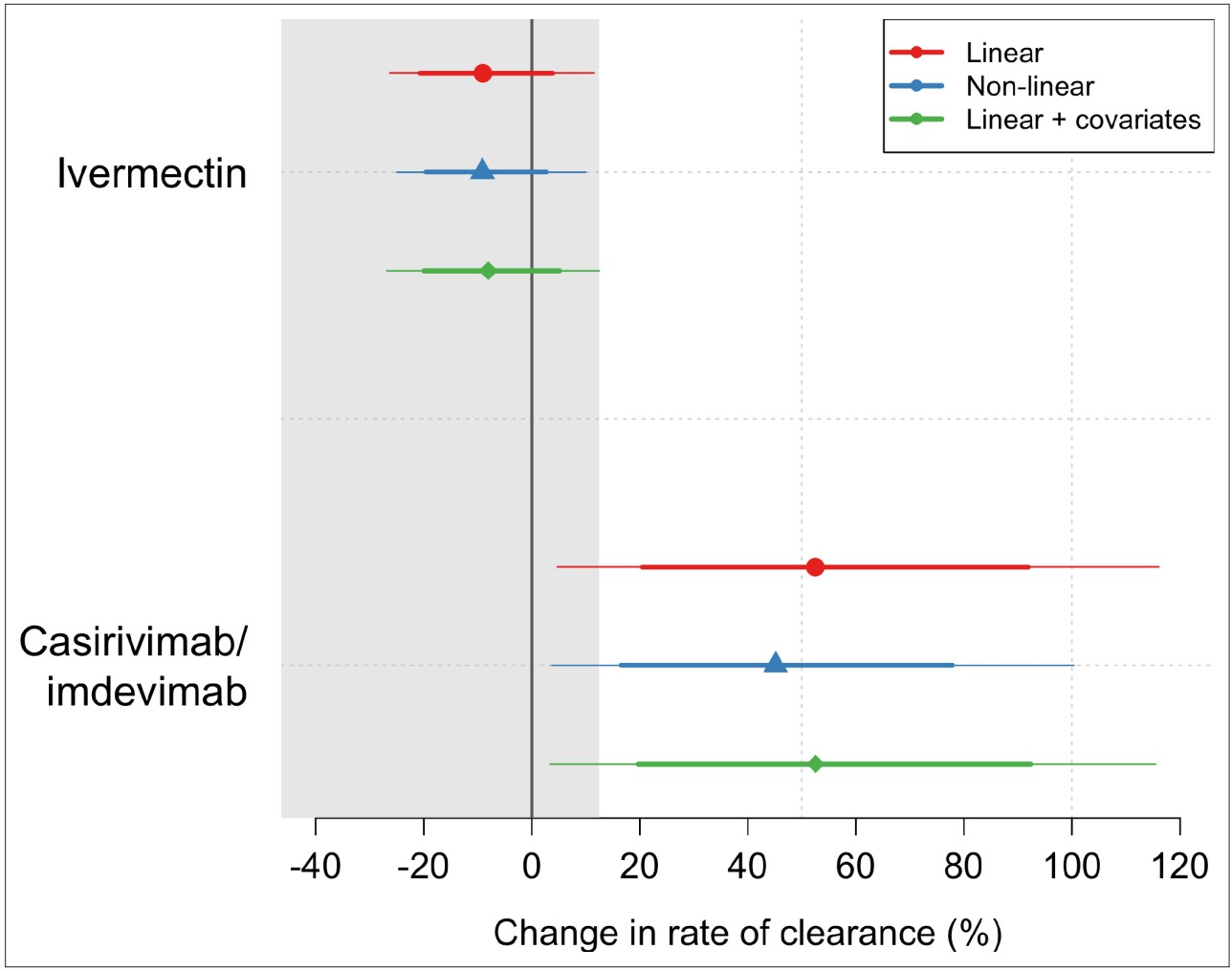

**Figure 4.** Treatment effects mean posterior estimates of the differences in the rate of viral clearance (thick dots) compared to the no study drug arm. 80% (thick lines) and 95% (thin lines) credible intervals under three hierarchical Bayesian models are shown. The gray area shows the futility zone (<12.5% increase in the rate of viral clearance). Results from three models are shown. The main model used to report effect estimates in the text is the linear model (red).

The online version of this article includes the following figure supplement(s) for figure 4:

**Figure supplement 1.** Covariate effects estimated for the main analytical linear model.

**Figure supplement 2.** Covariate effects on the intercept and slope.

**Figure supplement 3.** Treatment effect estimates for all nine models fit to the data.

The quantitative relationship between the antiviral effect and clinical response varies, as host factors and viral virulence are both important determinants of outcome. However, many of the studies showing improved clinical outcomes did so in high-risk, unvaccinated, and previously uninfected adult patients with SARS CoV-2 virus variants which are no longer prevalent. The magnitude of the effects measured is sensitive to study design and to temporal and epidemiological influences (*Sigal et al., 2022*). Direct comparisons between antiviral drugs using the clinical endpoints of the published phase III studies therefore leave much uncertainty.

This trial has several limitations. It set a futility threshold of <12.5% acceleration of viral clearance, as currently available specific antiviral therapies provide approximately 30–50% acceleration (*Jayk Bernal et al., 2022*; *Hammond et al., 2022*; *Robinson et al., 2022*). It does not exclude smaller antiviral effects, or benefit from a non-antiviral effect (e.g., an immunomodulatory effect). Whether a smaller antiviral effect would warrant recommendation in treatment is debatable, but it could still be very useful in prevention, which requires less potent viral suppression for a clinical benefit. It is

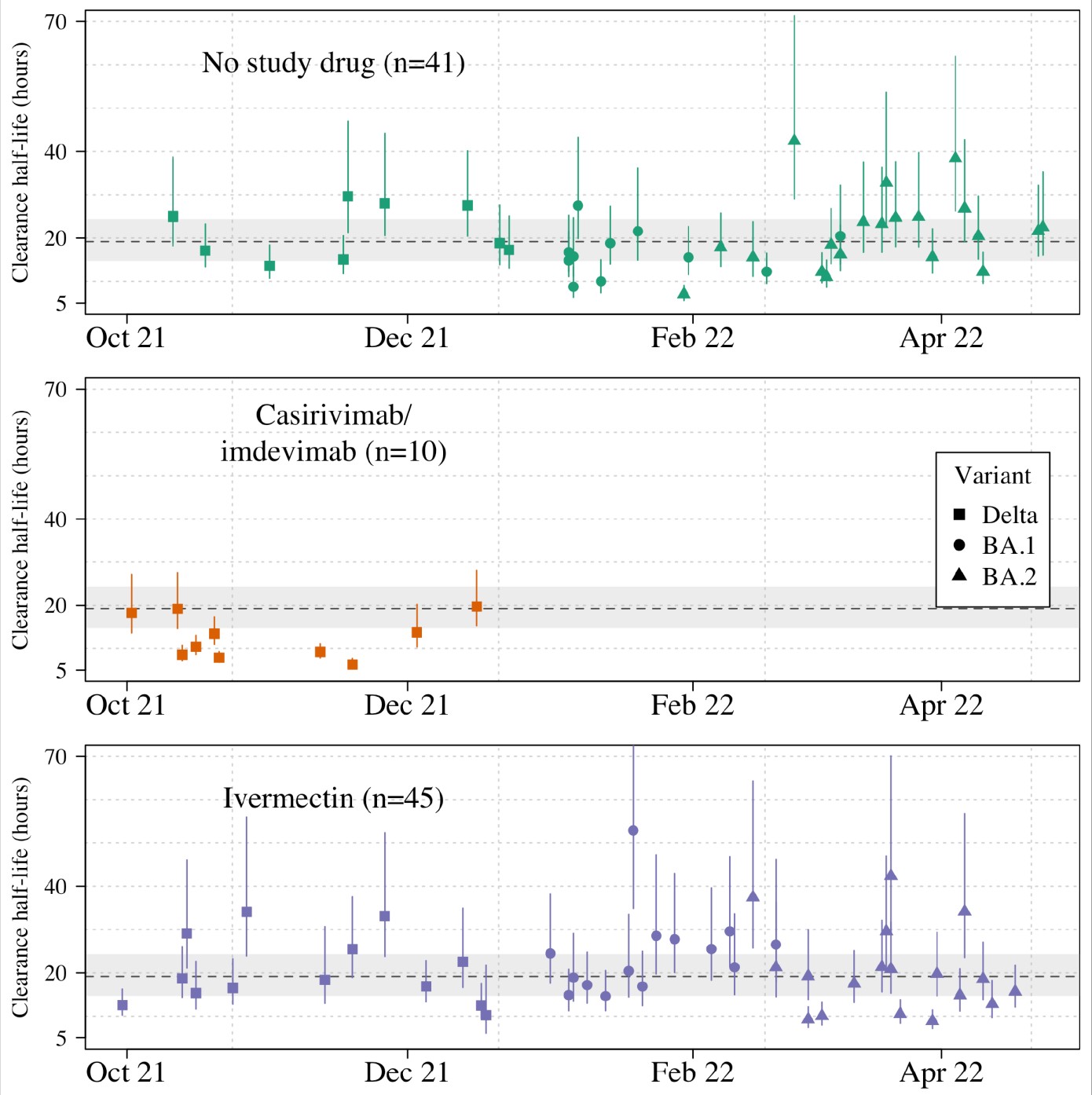

**Figure 5.** Individual patient virus clearance half-life estimates over time. The individual oropharyngeal virus clearance half-life mean posterior estimates with 95% credible intervals (lines) are shown (squares/circles/triangles corresponding to the virus genotype: temporarily Delta, Omicron BA.1, Omicron BA.2, respectively). The model estimated mean clearance half-life (95%CI) in untreated patients is shown by the grey line (dashed line-shaded area).

The online version of this article includes the following figure supplement(s) for figure 5:

**Figure supplement 1.** Estimated clearance half-lives in all patients in the mITT population (n=96), grouped by treatment arm and in order of decreasing rate of clearance.

**Figure supplement 2.** Individual fits to the serial qPCR data under the two main Bayesian hierarchical models (pink: linear model with RNase P adjustment; green: nonlinear model with RNase P adjustment).

also uncertain whether the daily sampling schedule is the optimal balance between statistical power and trial feasibility and acceptability. This could change as continued viral evolution and increasing vaccine coverage potentially affect viral clearance parameters. There is substantial variability in estimated serial viral loads (*Figure 5—figure supplement 2*). Whether variability can be reduced by adjusting for extracellular fluid content or different sampling techniques is uncertain. The viral qPCR measures viral genomes and does not distinguish between live (potentially transmissible) and dead virus. Finally, although not primarily a safety study, the lack of blinding compromises safety or tolerability assessments.

In summary, high-dose ivermectin did not have measurable antiviral activity in early symptomatic COVID-19. This study provides no support for the continued use of ivermectin in COVID-19. Efficient characterization and comparison of potential antiviral therapeutics in COVID-19 will be important for policy recommendations, particularly while cost and availability limit access. Use of the rate of oropharyngeal viral clearance as a metric for antiviral efficacy has applicability beyond COVID-19 including other respiratory illnesses, notably influenza (*Yu et al., 2010*), novel coronaviruses and future, as yet unknown, respiratory illnesses.

## Acknowledgements

The authors thank the patients with COVID-19 who volunteered to be part of this study. The authors thank the data safety and monitoring board (DSMB): Tim Peto, Andre Siqueira, and Panisadee Avirutnan, and the trial steering committee (TSC): Nathalie Strub-Wourgaft, Martin Llewelyn, Deborah Waller, and Attavit Asavisanu. The authors thank Sompob Saralamba and Tanaphum Wichaita for developing the RShiny randomization app and Mavuto Mukaka for statistical support. The authors also thank all the staff at the Clinical Trials Unit (CTU) at MORU. The authors are grateful to the PCR Expert group (Janjira Thaipadungpanit, Audrey Dubot-Pérès, Clare Ling and Elizabeth Batty), Thermo Fisher for their excellent support with this project, and all the hospital staff at the Hospital of Tropical Diseases (HTD), Bangplee (BP) and Vajira (VJ) hospitals, as well as those involved in sample processing in MORU and the processing and analysis at the Faculty of Tropical Medicine, molecular genetics laboratory, and also the MORU Clinical Trials Support Group (CTSG) for data management and logistics, and the purchasing, administration and support staff at MORU. A CC BY, or equivalent licence, is applied to any author accepted manuscript arising from this submission, in accordance with the grant's open access conditions.

## Additional information

### Competing interests

Mauro Martins Teixeira: Reviewing editor, eLife. The other authors declare that no competing interests exist.

### Funding

| Funder | Grant reference number | Author |
| --- | --- | --- |
| Wellcome Trust | 223195/Z/21/Z | William HK Schilling |

The funders had no role in study design, data collection and interpretation, or the decision to submit the work for publication. For the purpose of Open Access, the authors have applied a CC BY public copyright license to any Author Accepted Manuscript version arising from this submission.

### Author contributions

William HK Schilling, Supervision, Funding acquisition, Investigation, Methodology, Writing – original draft, Project administration, Writing – review and editing; Podjanee Jittamala, Investigation, Methodology, Project administration, Writing – review and editing; James A Watson, Data curation, Formal analysis, Funding acquisition, Visualization, Writing – original draft, Writing – review and editing; Maneerat Ekkapongpisit, Pongtorn Hanboonkunupakarn, Investigation, Project administration; Tanaya Siripoon, Thundon Ngamprasertchai, Kittiyod Poovorawan, Mauro Martins Teixeira, Investigation,

Writing – review and editing; Viravarn Luvira, Borimas Hanboonkunupakarn, Mallika Imwong, Sasithon Pukrittayakamee, Arjen M Dondorp, Watcharapong Piyaphanee, Weerapong Phumratanaprapin, Supervision, Investigation, Writing – review and editing; Sasithorn Pongwilai, Project administration, Writing – review and editing; Cintia Cruz, Funding acquisition, Investigation, Project administration, Writing – review and editing; James J Callery, Funding acquisition, Methodology, Writing – review and editing; Simon Boyd, Investigation, Methodology, Writing – original draft, Project administration, Writing – review and editing; Varaporn Kruabkontho, Jaruwan Tubprasert, Project administration; Thatsanun Ngernseng, Data curation; Mohammad Yazid Abdad, Investigation, Project administration, Writing – review and editing; Nattaporn Piaraksa, Kanokon Suwannasin, Janjira Thaipadungpanit, Joel Tarning, Vasin Chotivanich, Chunlanee Sangketchon, Kesinee Chotivanich, Investigation; Sakol Sookprome, Stuart Blacksell, Wiroj Ruksakul, Supervision, Investigation; Walter RJ Taylor, Investigation, Methodology; Nicholas PJ Day, Funding acquisition, Investigation, Writing – review and editing; Nicholas J White, Conceptualization, Formal analysis, Supervision, Funding acquisition, Investigation, Methodology, Writing – original draft, Writing – review and editing

## Author ORCIDs
William HK Schilling ⓘ http://orcid.org/0000-0002-6328-8748
James A Watson ⓘ http://orcid.org/0000-0001-5524-0325
Viravarn Luvira ⓘ http://orcid.org/0000-0001-9270-3720
Cintia Cruz ⓘ http://orcid.org/0000-0001-8393-8536
James J Callery ⓘ http://orcid.org/0000-0002-3218-2166
Janjira Thaipadungpanit ⓘ http://orcid.org/0000-0002-6184-3381
Joel Tarning ⓘ http://orcid.org/0000-0003-4566-4030
Mauro Martins Teixeira ⓘ http://orcid.org/0000-0002-6944-3008
Arjen M Dondorp ⓘ http://orcid.org/0000-0001-5190-2395
Nicholas PJ Day ⓘ http://orcid.org/0000-0003-2309-1171
Nicholas J White ⓘ http://orcid.org/0000-0002-1897-1978

## Ethics
Clinical trial registration NCT05041907.
Human subjects: The trial was approved by local and national research ethics boards in Thailand (Faculty of Tropical Medicine Ethics Committee, Mahidol University, FTMEC Ref: TMEC 21-058) and the Central Research Ethics Committee (CREC, Bangkok, Thailand, CREC Ref: CREC048/64BP-MED34) and by the Oxford University Tropical Research Ethics Committee (OxTREC, Oxford, UK, OxTREC Ref: 24-21). All patients provided fully informed written consent.

## Decision letter and Author response
Decision letter https://doi.org/10.7554/eLife.83201.sa1
Author response https://doi.org/10.7554/eLife.83201.sa2

# Additional files

## Supplementary files
• MDAR checklist
• Supplementary file 1. Summary of adverse events (grade 3 and above).
• Supplementary file 2. Summary of Serious Adverse Events.

## Data availability
All code and data are openly accessible via GitHub: https://github.com/jwatowatson/PLATCOV-Ivermectin (copy archived at swh:1:rev:9cd724266ab4f7eab09c5d2a7fd50dac52782957). Sequencing data have been deposited in GISAID.

The following dataset was generated:

| Author(s) | Year | Dataset title | Dataset URL | Database and Identifier |
| --- | --- | --- | --- | --- |
| Watson J | 2022 | PLATCOV Ivermectin | https://github.com/jwatowatson/PLATCOV-Ivermectin | github, PLATCOV-Ivermectin |

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

# Appendix 1

## Virus variant determination

Viral genetic variants were identified using real-time PCR genotyping with the TaqMan SARS-CoV-2 Mutation Panel. The variants circulating in Thailand from September 30, 2021 to April 18, 2022 were Delta (B.1.617.2), Omicron BA.1 (B.1.1.529), and Omicron BA.2 subvariants (B.1.1.529). All samples were tested for four canonical mutations of the circulating variants. Those with mutation S.T19R. ACA.AGA were designated Delta (B.1.617.2), if no other mutations in the panel were identified. Those with mutation S.Q493R.CAA.CGA were designated as Omicron BA.1 (B.1.1.529) if no other mutations in the panel were identified. Those with mutations S.Q493R.CA.A.CGA, S.T376A.ACT. GCT, and S.V213G.GTG.GGG were designated Omicron BA.2 (B.1.1.529), if S.T19R.ACA.AGA was absent.

Identifications were confirmed using Whole-Genome Sequencing as below:

The sequencing method carried out in this experiment follows the 'PCR tiling of SARS-CoV-2 virus with rapid barcoding and Midnight RT PCR Expansion' provided by Oxford Nanopore Technology (Oxford, UK) developed based on a protocol by the ARTIC network group. Library preparation process started with reverse transcription, which consists of mixing the purified viral RNA with LunaScript RT SuperMix and incubating the mixtures in a thermal cycler. DNA fragments to be used in the assembly process were amplified by PCR using Midnight primer set (V3) and attached with barcodes from Rapid Barcode Plate (RB96). The mixtures from each sample were pooled together, cleaned with AMPure XP Beads (AXP), and attached with Rapid Adapter F (RAP F). The prepared DNA fragments were then loaded into a primed flow cell (FLO-MIN106) and sequenced on GridION MK1 system.

## Viral genome assembly and classification

The output sequencing data (.fast5) from MinKNOW software was base-called with Guppy software using the High Accuracy (HAC) model to generate nucleotide sequence data for each fragment (reads) in the fastq format. These base-called data were then processed through the established workflow wf-artic on EPI2ME software to be assembled into consensus sequences. Only reads with average Phred Quality (Q) score above 9 and minimum and maximum length of 250 and 1500 bps were used in the assembly process. The consensus sequences were then classified using the Pangolin tool (4.1.1) and Pangolin data set (v1.14).

## Appendix 2

### Safety and ivermectin visual side effects
Adverse events and serious adverse events
See supplementary files for: *Supplementary file 1* and *Supplementary file 2*.

### Ivermectin visual side effects
Six patients receiving ivermectin complained of visual disturbances although none of these met the predefined AE criteria of ≥3. Ivermectin at high doses is known to cause transient visual side effects (*Navarro et al., 2020*; *Smit et al., 2018*). Six patients in the trial who experienced transient visual symptoms were all reviewed by a qualified medical physician, in discussion with the PLATCOV Safety Team and the DSMB and were not considered to be a safety concern for the participants. As these side effects have been well documented previously (*Navarro et al., 2020*; *Smit et al., 2018*), were transient and caused no lasting damage, randomization into the ivermectin arm was not halted for safety reasons (although individual patients did switch to alternative treatments). Of note, two other participants who did not receive ivermectin also experienced similar transient visual changes. These occurred in a participant who received no study drug and another participant who received remdesivir. These cases were discussed with the DSMB committee who were in agreement with the study team's decision.

| Case | History |
|---|---|
| 1 | On discharge, the participant reported they had been having episodes of unilateral dark gray/black shadowing of the lower half of the right eye's visual field after four doses of ivermectin. Episodes lasted 5 s and only happened once a day, several hours after receiving ivermectin. They were reviewed by an ophthalmologist whose examination of the patient was normal. Symptoms resolved following cessation of ivermectin. |
| 2 | The participant experienced visual changes in both eyes after three doses of ivermectin. There was sudden peripheral blurring/fogging lasting 2 s which self-resolved. Later, widespread black dots developed bilaterally, again lasting 2 s and which again self-resolved. Bedside examination was normal. Ivermectin was stopped at the patient's request. |
| 3 | The participant developed a headache, then later blurred vision bilaterally after one dose of ivermectin. Three hours after dosing, there was a bilateral throbbing pain in of the head (no aura) which was relieved with paracetamol and sleep. The next morning (18 hr post-administration), there was bilateral fogging of both eyes. Symptoms lasted 30 s to 1 min and self-resolved. There was no pain, floaters or flashing, and it was unrelated to position. Examination was normal. Ivermectin was discontinued and there were no further issues. |
| 4 | The participant informed the clinical staff on day 5 that they had been experiencing daily episodes of blurred vision (appearance of clouding) since being started on ivermectin. This was initially in the left eye but then later in both eyes. The episodes lasted about 10 min. They were worse on lying down and the timing was related to the administration of ivermectin. They received five doses of ivermectin in total. Eye examination was normal. Episodes spontaneously resolved following cessation of ivermectin. |
| 5 | The participant informed the ward staff on day 5 that they have been having daily transient visual changes since being given ivermectin (symptoms similar to case 4, both participants were on the study ward together). Symptoms consisted of short episodes of visual blurring in both eyes with some associated dizziness which improved without treatment. |
| 6 | The participant informed the study team at the day 28 follow-up that they had been experiencing intermittent blurring of vision from day 3 of the study. Symptoms had subsequently begun to resolve on discharge (day 7). The participant was offered an ophthalmology appointment but declined further follow-up. |

## Appendix 3

### Randomization

The randomization sheets were generated by the trial statistician (James Watson).

All new randomization sheets and all updates of existing randomization sheets were done using a prewritten R script which was stored on the randomization Dropbox folder owner is MORU, under custodianship of the head of MORU IT; this file is a full 'Professional' version with history recorded and only the trial statistician and head of IT had access. The file took the following inputs:

- Site codes (e.g., 'th001') for which to generate randomization sheets;
- The set of arms available for randomization in that site;
- The number of arm repeats per block (i.e., set to the minimum integer such that in each block there is an integer number for each arm);
- The randomization data file from each site (which has the patient numbers for subjects already randomized) named data-XXX.csv (where XXX is the site code), if this does not yet exist a blank csv (headers only) is generated.

This R script is run every time a new site becomes active and every time the set of available arms changes. The output is a csv file named rand-XXX.csv (where XXX is the site code). This overwrites the preexisting file (which can be retrieved from the Dropbox version history). Each time the randomization script is run, this is recorded on a log file.

The randomization is done according to the following constraints:

- Blocks of 2*number of available arms;
- Additional 'fuzziness' by swapping one patient allocation per block at random (this can be swapped for any of the available arms)—this avoids knowing which arm the last patient per block will receive.

Each time an authorized member of the study team logs onto the web-app this is logged (timestamp and username).

Each time a new patient is randomized this is logged on to the file data-XXX.csv (where XXX is the site code) with the following information:

- Subject number
- Screening number
- Age
- Sex
- Member of study team username
- Timestamp

At the start of the trial (September 30, 2021), randomization to casirivimab/imdevimab was set at 10% (positive control), and the other arms had equal uniform randomization ratios of 22.5%. This was changed on October 30, 2021, so that all arms had equal randomization ratios. This explains why there are slightly fewer than expected patients randomized to casirivimab/imdevimab.

# Appendix 4

## Statistical analysis

The primary analysis consists of fitting Bayesian hierarchical (mixed effects) linear models to the serial $\log_{10}$ viral load data up until day 7 (day 14 data were not used). All models encode residual error as a t-distribution with degrees of freedom estimated from the data. The t-distribution was chosen for robustness as the residual error is clearly non-Gaussian. The t-distribution error model also makes the model robust against model mis-specification (particularly for the linear models) (*Lange et al., 1989*). All models include correlated individual random effect terms for both the intercept (baseline viral load) and the slope. All changes to the slope are defined as multiplicative changes on the log scale (a value of 0 equals no change).

The treatment effect is defined as the proportional change (expressed as a multiplicative term) in the population slope of the daily change in $\log_{10}$ viral load. The data are modeled on the $\log_{10}$ copies per mL scale, after conversion from Ct values using the standard curve generated from the 12 control concentrations ( samples with known viral densities) from each 96-well plate. The standard curve transformation is done by fitting a linear mixed effects model (random slope and random intercept for each plate) to the control data: regressing the Ct values on the known log viral densities. This borrows information across plates and allows for batch effects.

For all models, we adjusted the intercept and slope for the enrolling site (three sites in total, the reference site is the Hospital of Tropical Diseases which recruited >90% of patients) and for the variant called (Delta is reference: BA.1 and BA.2 are the two alternatives). A subset of models also adjusted the slopes and intercepts for:

- Age
- Number of vaccine doses
- Result of serology antibody test
- Days since symptom onset

All models except model 1 adjust for human RNase P (proxy for the number of human cells in the sample).

In total, we fit nine separate models:

1. Model 1 is linear with no RNase P adjustment; adjustment for site & variant; weakly informative priors (WIP)
2. Model 2 is linear with RNase P adjustment; adjustment for site & variant; WIP. This is the main model used to report treatment effects.
3. Model 3 is non-linear; RNase P adjustment; adjustment for site & variant; WIP
4. Model 4 is linear with RNase P adjustment; adjustment for site & variant; non-informative priors (NIP)
5. Model 5 is nonlinear with RNase P adjustment; adjustment for site & variant; NIP
6. Model 6 is linear with RNase P adjustment; full covariate adjustment; WIP
7. Model 7 is nonlinear with RNase P adjustment; full covariate adjustment; WIP
8. Model 8 is linear with RNase P adjustment; full covariate adjustment; NIP
9. Model 9 is nonlinear with RNase P adjustment; full covariate adjustment; NIP

Model 1 is a base model without RNase P adjustment; models 2–9, all have RNase P adjustment and are all combinations of linear and nonlinear models, with or without full covariate adjustment; and with either weakly informative priors or non-informative priors.

We compared model fits using the *loo* (approximate leave-one-out cross validation) package.

The statistical analysis plan provides a detailed overview of the model structures. Comparison of treatment effect under all nine models is given in *Figure 4—figure supplement 3*. All data, models, and analytical output are on the linked GitHub repository: https://github.com/jwatowatson/ PLATCOV-Ivermectin. This includes all data used in the analysis for full reproducibility of the results.

