## [Editor Report]

This valuable clinical trial demonstrated in a convincing fashion that ivermectin does not increase the rate of SARS-CoV-2 clearance from the oral compartment. This work will be of interest to clinicians and virologists and further demonstrates the use of viral clearance rate as a possible surrogate marker for SARS-CoV-2 antiviral trials.

---

## [Decision Letter]

**Decision letter after peer review:**

Thank you for submitting your article "Pharmacometrics of high dose ivermectin in early COVID-19: an open label, randomized, controlled adaptive platform trial (PLATCOV)" for consideration by *eLife*. Your article has been reviewed by 3 peer reviewers, including Joshua T Schiffer as Reviewing Editor and Reviewer #1, and the evaluation has been overseen by Neil Ferguson as the Senior Editor. The following individuals involved in the review of your submission have agreed to reveal their identity: Rachel Bender Ignacio (Reviewer #2); David Boulware (Reviewer #3).

Essential revisions:

1) Describe existing papers that have established virologic surrogate endpoints for clinical outcomes, and the particular challenges associated with achieving this goal for SARS-CoV-2 described by reviewers 1 and 2.

2) Clarify the unblinding rules.

3) Describe in more detail how swabs from different sires are processed.

4) Describe allocation into the study arms.

5) Describe how the modeling accounts for viral rebound and undetectable viral load measures.

6) Provide more precise language to discriminate clinical significance from statistical significance.

*Reviewer #2 (Recommendations for the authors):*

This reviewer wishes to compliment the authors on this highly important collaboration, which is bringing innovative clinical science of global importance, with translatability across many settings, whether in high-income or low and middle-income countries with limited access to therapeutics. For future work that arises from this collaboration, we hope that the authors will consider the power and funding dynamics and authorship order in a way that further supports colleagues from the local institutions in pursuing first/last authorship in future papers. The included reference is with respect to collaborations between European and N. American research institutions with African institutions but challenges the paradigm of middle-only authorship from local institutions, and may be applicable in other settings in which funding comes with collaborators from outside countries.

Hedt-Gauthier BL, Jeufack HM, Neufeld NH, et al. Stuck in the middle: a systematic review of authorship in collaborative health research in Africa, 2014-2016. BMJ Global Health 2019;4:e001853. doi:10.1136/bmjgh-2019-001853

---

## [Author Response]

Essential revisions:1) Describe existing papers that have established virologic surrogate endpoints for clinical outcomes, and the particular challenges associated with achieving this goal for SARS-CoV-2 described by reviewers 1 and 2.

Virologic surrogate end-points have not been well established in COVID-19, unlike for other viral illnesses, where viral load has been shown to be a good surrogate for clinical endpoints (e.g. CMV infections in solid organ transplants, or HIV or HCV infections). The main papers describing the efficacy of the specific monoclonal antibodies (notably casivirimab and imdevimab) and of molnupiravir and nirmatrelvir (which we have referenced [1,4,5]) have noted that viral clearance was accelerated with these interventions, each of which was associated with therapeutic benefit (prevention of hospitalisation). Viral clearance was used as the pharmacodynamic end-point in the molnupiravir dose finding studies (Ref 1 below). However, in each of these studies, viral loads were assessed infrequently, which substantially reduced the precision of the estimate of viral clearance rate (rates were not specifically measured in these studies). In contrast, the PINETREE study [16] of remdesivir did not find a relationship between therapeutic efficacy (prevention of disease progression) and viral clearance, although again sampling was infrequent (increasing the possibility of a type 2 error). In contrast, using the more sensitive and precise methodology described in this paper, we have shown clearly that remdesivir does accelerate viral clearance, as would be expected from an efficacious antiviral drug. We have now made these points in the discussion (lines 377-382):

“Using less frequent nasopharyngeal sampling in larger numbers of patients, clinical trials of monoclonal antibodies, molnupiravir and ritonavir-boosted nirmatrelvir, have each shown that accelerated viral clearance is associated with improved clinical outcomes [1,4,5]. These data suggest reduction in viral load could be used as a surrogate of clinical outcome in COVID-19. In contrast the PINETREE study, which showed that remdesivir significantly reduced disease progression in COVID-19, did not find an association between viral clearance and therapeutic benefit. This seemed to refute the usefulness of viral clearance rates as a surrogate for rates of clinical recovery . However, the infrequent sampling in all these studies substantially reduced the precision of the viral clearance estimates (and thus increased the risk of type 2 errors). Using the frequent sampling employed in the PLATCOV study, we have shown recently that remdesivir does accelerate SARS-CoV-2 viral clearance [17], as would be expected from an efficacious antiviral drug. This is consistent with therapeutic responses in other viral infections [18, 19]. Taken together the weight of evidence suggests that accelerated viral clearance does reflect therapeutic efficacy”

2) Clarify the unblinding rules.

We have added text explaining how the interim analyses were performed (frequency). The interim analyses were sent to the DSMB who reviewed the data and analysis and ratified the decision to stop a treatment arm once a stopping rule was met.

3) Describe in more detail how swabs from different sires are processed.

Swabs from all sites are taken and processed the same way. Identical swabs and viral transport media (VTM) were used in each site, and each person doing the swabbing was trained to use a standardised swabbing technique. Lines 161-165:

“A Thermo Fisher MicroTest flocked swab was rotated against the tonsil through 360^o^ four times and placed in Thermo Fisher M4RT viral transport medium (3mL). On subsequent days (day 1 to day 7 and then after discharge on day 14), a single swab was taken from each tonsil (left and right, total of 2 swabs). Swabs were transferred separately at 4-8^o^C, aliquoted and then frozen at -80^o^C within 48hrs.”

The VTM was frozen at sites and then transported at -80^o^C to the central laboratory in the Faculty of Tropical Medicine (FTM), Mahidol University, Bangkok for further testing, whereby all the onward processing and testing was identical.

The only difference in the processing between sites was that the swab VTM in FTM was aliquoted prior to freezing at -80^o^C, whereas the swab VTM in other sites was frozen and aliquoted after defrosting, immediately prior to testing. As all samples had the same number of freeze-thaws prior to testing (i.e. one), the processing can be considered to be the same.

4) Describe allocation into the study arms.

The randomization was uniform across intervention arms with a minimum of 20% of patients allocated to no study drug. Lines 143-147:

“The no study drug arm comprised a minimum proportion of 20% and uniform randomization ratios were then applied across the treatment arms. For example, for five intervention arms plusthe no study drug arm, 20% of patients would be randomized to no study drug and 16% to each of the 5 interventions. Additional details on the randomization are provided in the Appendices.”

5) Describe how the modeling accounts for viral rebound and undetectable viral load measures.

The models intentionally do not account for viral rebound which, if it occurs, usually does so after one week.

Undetectable viral loads are formally accounted for by treating them as left censored (the model likelihood is equal to cumulative density function).

We have added to the Methods. Lines 185-191:

“Viral loads below the lower limit of quantification (Ct values >40) were treated as left-censored under the model with a known censoring value. The PCRs were done on 96 well plates, each of which included 12 standards of known viral density. The Ct values from the patient swabs were then converted to copies per mL under standard curves estimated using the control data from all available plates (a mixed-effects linear regression model with a random slope and intercept for each plate; each plate therefore has a slightly different left-censoring value).”

6) Provide more precise language to discriminate clinical significance from statistical significance.

We have changed this throughout the text.

Reviewer #2 (Recommendations for the authors):This reviewer wishes to compliment the authors on this highly important collaboration, which is bringing innovative clinical science of global importance, with translatability across many settings, whether in high-income or low and middle-income countries with limited access to therapeutics. For future work that arises from this collaboration, we hope that the authors will consider the power and funding dynamics and authorship order in a way that further supports colleagues from the local institutions in pursuing first/last authorship in future papers. The included reference is with respect to collaborations between European and N. American research institutions with African institutions but challenges the paradigm of middle-only authorship from local institutions, and may be applicable in other settings in which funding comes with collaborators from outside countries.Hedt-Gauthier BL, Jeufack HM, Neufeld NH, et al. Stuck in the middle: a systematic review of authorship in collaborative health research in Africa, 2014-2016. BMJ Global Health 2019;4:e001853. doi:10.1136/bmjgh-2019-001853

We thank the reviewer for raising this important point. Many individuals have contributed time and effort to set up this platform trial and we are lucky in that it will generate multiple important publications, which will continue to reflect the contributions of those involved in the study.

Reference

1. Khoo SH, Fitzgerald R, Fletcher T, Ewings S, Jaki T, Lyon R, Downs N, Walker L, Tansley-Hancock O, Greenhalf W, Woods C, Reynolds H, Marwood E, Mozgunov P, Adams E, Bullock K, Holman W, Bula MD, Gibney JL, Saunders G, Corkhill A, Hale C, Thorne K, Chiong J, Condie S, Pertinez H, Painter W, Wrixon E, Johnson L, Yeats S, Mallard K, Radford M, Fines K, Shaw V, Owen A, Lalloo DG, Jacobs M, Griffiths G. Optimal dose and safety of molnupiravir in patients with early SARS-CoV-2: a Phase I, open-label, dose-escalating, randomized controlled study. J Antimicrob Chemother. 2021 Nov 12;76(12):3286-3295. doi: 10.1093/jac/dkab318. PMID: 34450619; PMCID: PMC8598307.